# The Effect of Combining Millet and Corn Straw as Source Forage for Beef Cattle Diets on Ruminal Degradability and Fungal Community

**DOI:** 10.3390/ani13040548

**Published:** 2023-02-04

**Authors:** Yaoyi Tong, Jincai Wu, Wenwei Guo, Zhimin Yang, Haocheng Wang, Hongkai Liu, Yong Gao, Maohong Sun, Chunwang Yue

**Affiliations:** 1College of Animal Science and Technology, Hebei North University, Zhangjiakou 075051, China; 2Zhangjiakou Academy of Agricultural Sciences, Zhangjiakou 075000, China; 3Three Steps Fattening Feed R&D Center Co., Ltd., Beijing 102206, China

**Keywords:** millet straw, corn straw, ruminal degradability, fungal community

## Abstract

**Simple Summary:**

Millet straw and corn straw as roughages for the ruminant diet have broad application prospects. A variety of roughage combinations affect the ruminal degradability and rumen fungal community. The effects of roughage mixture on the rumen of beef cattle are little understood. This research focuses on the impact of straw on beef cattle feed nutrition, as well as different ratios of millet straw and corn straw diets on ruminal degradability and rumen fungal community. The results proved the feeding effect of different proportions of millet straw and corn straw combinations in beef cattle.

**Abstract:**

Three ruminal cannulated Simmental crossbreed bulls (approximately 3 years of age and with 380 ± 20 kg live weight at initiation of the experiment) were used in a 3 × 3 Latin square experiment in order to determine the effects of the treatments on ruminal pH and degradability of nutrients, as well as the rumen fungal community. The experimental periods were 21 d, with 18 d of adjustment to the respective dietary treatments and 3 d of sample collection. Treatments consisted of a basal diet containing a 47.11% composition of two sources of forage as follows: (1) 100% millet straw (MILLSTR), (2) 50:50 millet straw and corn straw (COMB), and (3) 100% corn straw (CORNSTR). Dry matter (DM), crude protein (CP), neutral detergent fiber (NDF), and acid detergent fiber (ADF) were tested for ruminal degradability using the nylon bag method, which was incubated for 6, 12, 24, 36, 48, and 72 h, and rumen fungal community in rumen fluid was determined by high-throughput gene sequencing technology. Ruminal pH was not affected by treatments. At 72 h, compared to MILLSTR, DM degradability of CORNSTR was 4.8% greater (*p* < 0.05), but when corn was combined with millet straw, the difference in DM degradability was 9.4%. During the first 24 h, degradability of CP was lower for CORNSTR, intermediate for MILLSTR, and higher for COMB. However, at 72 h, MILLSTR and COMB had a similar CP degradability value, staying greater than the CP degradability value of the CORNSTR treatment. Compared to MILLSTR, the rumen degradability of NDF was greater for CORNSTR and intermediate for the COMB. There was a greater degradability for ADF in CORNSTR, intermediate for COMB, and lower for MILLSTR. In all treatments, *Ascomycota* and *Basidiomycota* were dominant flora. Abundance of *Basidiomycota* in the group COMB was higher (*p* < 0.05) than that in the group CORNSTR at 12 h. Relative to the fungal genus level, the *Thelebolus, Cladosporium*, and *Meyerozyma* were the dominant fungus, and the abundance of *Meyerozyma* in COMB and CORNSTR were greater (*p* < 0.05) than MILLSTR at 12, 24, and 36 h of incubation. In conclusion, it is suggested to feed beef cattle with different proportions of millet straw and corn straw combinations.

## 1. Introduction

There were correlation effects among different feed combinations. Different combinations of diets can save forage resources, such as the sweet stalklage and wheat straw that can be used as an alternative to silage at a later stage of fattening beef cattle [1]. Different dietary combinations have positive effects on the fattening performance and the production performance of ruminants. The combination of cactus pear with elephant grass or sorghum silage in the diet influenced and increased the fatty acid profile [2]. In addition, the replacement of half of the corn straw with millet straw could improve the growth performance and the blood biochemistry metabolism capability of fattening lambs [3]. Furthermore, the dietary compositions, additive content, and type of additives affected the rate of rumen degradation and microbial community composition. The 50% proportion of orchardgrass and alfalfa as roughage is beneficial to the growth of rumen microorganisms in lactating dairy cows and does not affect nutrient digestion and rumen fermentation; it is also increased the rate of passage of small orchardgrass particles and the rate of DM and NDF digestion and nutrient intake [4]. The in vitro degradation rate of silage prepared by mixing sweet sorghum and alfalfa reached its peak when the sweet sorghum–alfalfa ratio reached 25:75 [5].

Fiber degradation in the rumen of ruminants is accomplished by the synergistic action of bacteria, fungi, and ciliates [6]. Trinci points out that the proportion of fungi in total rumen microorganisms was affected by dietary fiber content. When the amount of starch or grain was higher, the amount of fungus in the rumen was generally lower; on the contrary, when the fiber was higher, the fungus was higher [7]. According to Akin, there were more rumen fungi in sheep-fed sulfur-rich herbage than in sheep-fed sulfur-deficient herbage; even not-at-all sulfur-rich herbage might be used to encourage the formation of rumen fungi in animals [8]. The pseudoroot structure of fungi has significant penetration and fiber disintegration abilities, which makes the fiber tissue loose and amenable to degradation by other microorganisms, primarily in the form of erosion. As a result, it may break down lignified fibers, especially roughage, that bacteria and ciliates are unable to break down [9]. Most studies in the field of the rumen microbial community have focused on the bacterial community, but this study sought to analyze rumen activity from a fungal community perspective.

Forages are an essential component of ruminant diets for rumen function. Corn straw and millet straw have high utilization values for ruminants, which can relieve the pressure of shortages of high-quality forage grass and provide support for the development of new forage resources and the creation of characteristic ruminant products. The objective of this study was to investigate the effect of different millet and corn straw percentages in beef cattle diets on ruminal degradability and the rumen fungal community, using the nylon bag method and high-throughput gene sequencing technology, to provide a theoretical basis for the use of roughage combinations in beef cattle production.

## 2. Materials and Methods

All experimental procedures and animal experiments were performed following the guidelines of the Ethics Committee. This study was approved by the Institutional Animal Care and Use Committee of the Hebei North University (HBNU211030042087).

### 2.1. Animals, Experimental Design and Diets

Three ruminal cannulated Simmental crossbreed bulls (approximately 3 years of age and with 380 ± 20 kg live weight at initiation of the experiment) were selected as experimental animals. Treatments consisted of a basal diet containing a 47.11% composition of two sources of forage, as follows: (1) 100% millet straw (MILLSTR), (2) 50:50 millet straw and corn straw (COMB), and (3) 100% corn straw (CORNSTR). The experimental animals were used in a 3 × 3 Latin square experiment and the experimental periods were 21 d, with 18 d of adjustment to the respective dietary treatments and 3 d of sample collection. The simultaneous input and removal at different time points was used to collected the nylon bags placed in the rumen and incubated ruminal fluid which are incubated at 6, 12, 24, 36, 48, and 72 h. Two feedings each day at constant intake at 6:00 a.m. and 6:00 p.m. (feeding at the level of 2.2% of BW). The feed offer was in equal parts in morning and evening. Free water was available, and the barn was periodically cleaned and sanitized. Dietary and nutrient composition refer to NRC (2001) standards (Table 1, Table 2) [10]. The millet and corn straws were ground in a hammer mill with a 2.6 cm screen before inclusion in diets.

### 2.2. Ruminal Sample Collection, Sample Conservation, and Analyses

The diet samples were dried at 65.0 ℃ to constant weight in the oven and then placed at room temperature for 48 h. They were then ground in a small hammer mill and passed through a 1 mm sieve (40 mesh) before the chemical composition was determined.

Rumen fluid was collected by the suction device at above-mentioned incubated time after morning feeding on the first day of the collection sample period. The filtrate was filtered by four layers of gauze and placed in a 10 mL centrifuge tube for labeling, then stored in a refrigerator at −80 ℃ for analysis [11].

An electronic pH meter (Shanghai, China Zhiguang Instrument Co., Ltd. pHS-25) was lowered into the rumen by cannulas while being loaded with weights to continually record the dynamic variations in pH.

### 2.3. Procedures of In Situ Digestibility and Analyses

The 3.5 g crushed feed sample was precisely weighed and placed in a nylon bag (8 by 12 cm, mesh size of 38–40 µm). A 48 cm plastic rod was used, a 4 cm incision was made at one end of the plastic rod, and two nylon bags were fastened together with rubber bands. A small hole was ironed with a hot wire and pierced with an 80 cm nylon rope to make a ring, and a 15 cm round iron ring was fastened to the nylon rope. Before morning feeding, the ring was placed in the rumen and the fistula cover was closed. Each group of roughage was put into 3 nylon bags. After incubating for 6, 12, 24, 36, 48, and 72 h, one plastic tube was removed and cleaned with water immediately until the water was colorless. The washed nylon bag was immediately put into the oven at 65 ℃ to dry and determine the nutrient degradation rate [12].

### 2.4. Determination of Ruminal Degradability

The DM and CP were determined using an AOAC procedure (1990) [13]. The contents of NDF and ADF were determined using the methods by Van Soest et al. [14].

The parameters of the dynamic degradation model were calculated according to the following exponential equation [15]:$$p=a+b\left(1-e^{-\mathrm{ct}}\right)$$
where p is the real-time degradability of DM, CP, NDF, and ADF at time t; a is the rapidly degradable fraction (g/kg); b is the potentially degradable fraction (g/kg); c is the constant rate of degradation of b (%/h); and t is the time of incubation (h).

The effective degradability (ED) of nutrients was calculated according to the following equation [15]:$$\mathrm{ED}=a+\frac{\mathrm{bc}}{\mathrm{kp}+c}$$
where a, b, and c are the same parameters as the previous equation; kp (%/h) is the rumen particle passage rate, and its value of 0.031%/h was calculated according to NRC (2001) [10].

### 2.5. Determining Ruminal Composition and Diversity

The microbial DNA was extracted using the E.Z.N.A.^®^ soil DNA Kit (Omega Bio-Tek, Norcross, GA, USA) according to the manufacturer’s protocols, and the integrity of DNA was evaluated by 1% agarose gel electrophoresis. The PCR reactions were conducted using the following program: 3 min of denaturation at 95 ℃; 27 cycles of 30 s at 95 ℃; 30 s for annealing at 55 ℃; 45 s for elongation at 72 ℃; and a final extension at 72 ℃ for 10 min. The amplification region of fungal 18S rDNA was ITS1-ITS2, and the primer sequence was ITS1F (CTTGGTCATTTAGAGGAAGTAA) and ITS2R (GCTGCGTTCTTCATCGATGC). DNA fragments were sequenced using the Illumina HiSeq 2500 sequencing platform at Beijing Baemai Biotechnology Co., Ltd. (Beijing, China) to obtain raw data.

Purified amplicons were equimolarly pooled and paired-end sequenced on an Illumina MiSeq platform (Illumina, San Diego, CA, USA) according to the standard protocols [16]. To obtain the original tag data, the original sequences were spliced using FLASH software (version 1.2.11). Trimmomatic software (version 0.33) was used to filter the raw tags obtained to obtain high-quality clean tag data. Following that, chimeric sequences were located and eliminated using UCHIME software (version 8.1) to provide useful tags. Using USEARCH (version 10.0), a clustering software, the tags were grouped into operational taxonomic units (OTUs) based on a 97% sequence similarity level [16]. The obtained OTUs were later used for taxonomic assignment. To obtain taxonomic classification at the phylum, class, order, family, and genus levels, representative sequences from each OTU were compared to the Silva (Release128) database. The relative abundances of fungal community and the diversity of Beta were calculated and determined using QIIME (2022.8) software [17,18]. In the end, the MOTHUR software (version 1.30) was used to calculate richness and diversity indices to compare fungal diversity among different additives.

### 2.6. Statistical Analysis

A one-way ANOVA in SPSS 24.0 (SPSS Inc., Chicago, IL, USA) was used to determine the significance of the degradation rate of various nutrients in the rumen. All the effects were tested for statistical significance (*p* < 0.05), and significant effects were reported in the tables. When significant differences were found (*p* < 0.05), the *t*-test was used to locate significant differences between the means. Similarity analysis (ANOSIM) was used to analyze the significance of PCoA analysis of rumen fungi among different main factors [19].

## 3. Results

### 3.1. Ruminal pH and Ruminal Degradability

#### 3.1.1. Ruminal pH

At the same time point, groups had no significant effect on rumen fluid pH value (*p* > 0.05) (Table 3).

#### 3.1.2. Ruminal DM Degradability

According to Table 4, the ruminal DM degradability of COMB was higher (*p* < 0.05) than MILLSTR and CORNSTR at 12 and 24 h, and the ruminal DM degradability of COMB and CORNSTR was higher (*p* < 0.05) than MILLSTR at 72 h, but there was no significant difference (*p* > 0.05) in effective degradability among the three groups.

#### 3.1.3. Ruminal CP Degradability

As shown in Table 5, we can obtain the following results: the ruminal CP degradability of MILLSTR was lower (*p* < 0.05) than COMB and higher (*p* < 0.05) than CORNSTR at 6, 12, and 24 h; MILLSTR and COMB were higher (*p* < 0.05) than CORNSTR at 36, 48, and 72 h; and the effective degradability of COMB was lower (*p* < 0.05) than MILLSTR and higher (*p* < 0.05) than CORNSTR.

#### 3.1.4. Ruminal NDF Degradability

Table 6 compares the summary statistics for ruminal NDF degradability. CORNSTR was higher (*p* < 0.05) than MILLSTR at 24 h and 72 h, and there was no significant difference (*p* > 0.05) in effective degradability among the three groups.

#### 3.1.5. Ruminal ADF Degradability

Table 7 below illustrates that the ruminal ADF degradability of COMB was lower (*p* < 0.05) than CORNSTR and higher (*p* < 0.05) than MILLSTR at 6 h, CORNSTR was higher (*p* < 0.05) than MILLSTR at 48 h and 72 h, and there was no significant difference (*p* > 0.05) in effective degradability among the three treatments.

### 3.2. Rumen Fungal Community Composition and Diversity

#### 3.2.1. The Species and Structural Diversity

The results of the fungus α-diversity analysis are shown in Figure 1. For all samples, the average coverage was >99%, which could accurately reflect the species and structural diversity of fungal community. The ACE, Simpson, Chao1, and Shannon indexes were not significantly different (*p* > 0.05) among the treatment groups.

#### 3.2.2. The Fungus Composition on Phylum Level

At the phylum level, the abundance of *Ascomycota* and *Basidiomycota* was the highest among all the phylum, and *Ascomycota* accounted for more than 60% of the total number (Figure 2). According to the statistical results in Table 8, there was no significant difference (*p* > 0.05) in *Ascomycota* in the three treatments at the same incubation time, while *Basidiomycota* in COMB was higher (*p* < 0.05) than that in CORNSTR at 12 h.

#### 3.2.3. The Fungus Composition on Genus Level

At the genus level, *Thelebolus*, *Cladosporium* and *Meyerozyma* had the highest abundance among all fungus phyla (Figure 3). There was no significant difference (*p* > 0.05) in the number of *Thelebolus* and *Cladosporium* in three treatments; only *Meyerozyma* had significant difference in different incubated times (Table 9). Specifically, the number of *Meyerozyma* in MILLSTR was lower (*p* < 0.05) than that in CORNSTR at 12 h to 48 h; Additionally, the number of *Meyerozyma* in MILLSTR was lower (*p* < 0.05) than that in COMB at 12 h, 24 h, 36 h, and 72 h.

## 4. Discussion

This study analyzed the influence of various ruminant diets on ruminal nutrient degradability and changes in fungal communities in the rumen. Ruminal pH can directly reflect rumen fermentation level and rumen microbial activity in ruminants [20], and the results showed that the dynamic change of Ruminal pH reflects the change of organic acid quantity and saliva entry quantity in rumen digesta. In addition, the ruminal pH increased rapidly at 6 h, reached its maximum at 12 h, and then decreased slowly, showing a trend of decreasing first and then increasing. This might be caused by the rapid breakdown of concentrates in feed, especially soluble carbohydrates, the absorption of VFA by rumen epithelial cells, and the interaction of buffer in saliva and chyme outflow [21]. These findings may help us to understand the higher NDF content; physical stimulation increases rumination and chewing behavior, and stimulates saliva secretion, thus neutralizing the pH reduction. The three groups had no significant effect (*p* > 0.05) on ruminal pH at the same incubated times (Table 3). Furthermore, ruminal pHs were all within the safe range.

The ruminal DM and CP degradability of COMB mainly occurred within 24 h, which was the highest at 72 h. The slow degradation part c of COMB was higher (*p* < 0.05) than these of other treatments, which may be related to the combined effect produced by mixing corn straw and millet straw. On the one hand, dietary nutrient changes improved roughage digestibility. One the other, dietary protein and energy changes may improve the degradation ability of organic matter in roughages [22]. This result seems to be consistent with other research, which found the feeding effects of millet straw substituted for 50% of corn straw in fattening lambs [4]. The ruminal NDF and ADF degradability of MILLSTR was higher (*p* < 0.05) than those of other treatments, which might be related to the different lignin content in millet straw and corn straw, because lignin content is an important factor affecting the speed of rumen degradation rate of roughages [23,24]. In addition, corn straw had a looser fiber structure than millet straw, allowing rumen microbes to enter the roughage quickly and promote degradation. At the same time, phenolic substances released by lignocellulose metabolism may inhibit microbial activity [25]. As a result, the degradation rate of millet straw is superior to that of other groups, while that of millet straw is less ideal and will have an impact on the feeding of beef cattle. The results also showed that the degradation rates of NDF and ADF in the three treatments gradually increased, mainly after 12 h, and the degradation of roughages in each treatment was characterized by multiple stages [26]. This is consistent with the experimental results of Hoffman P C [27], indicating that the effective degradation of NDF and ADF of straw roughages needs to stay in the rumen for a certain period before the degradation rate accelerates, thus affecting the changes in the rumen microbial community. However, the findings of the current study do not match those of previous research, which could be due to the different breeds and feeding levels of the experimental animals. Compared with the results of Chen [3] on fattening lambs, it also had a positive effect on rumen performance of beef cattle.

Rumen fungi can partially destroy or weaken more resistant tissue through the herbaceous cuticle barrier, and they have a strong ability to penetrate and degrade plant fibers. As a result, they can degrade some lignified fibrous materials that cannot be decomposed by bacteria and ciliates [28]. Fungi adhere to segments of roughage in the rumen of ruminants, especially when fed diets high in fiber. The rumen fluid-collecting method mentioned above can still be used to acquire an accurate distribution of the amount of fungi present. At the fungal phylum level, after the experimental cattle were fed with different roughages, the dominant fungus were Ascomycota and Basidiomycota. According to Table 8, *Basidiomycota* in COMB was higher (*p* < 0.05) than that in CORNSTR at 12 h, and there was no significant difference (*p* > 0.05) at other incubation times, which seemed to indicate the adaptation of the rumen fungal community to dietary changes, but this was not persistent [29]. Contrary to the findings of Ishaq [30], a change in ph did have an impact on the variety and composition of the fungi community. This finding also suggested that the amount of crude fiber consumed could alter the fungi’s species. The distinction was that this paper focused mostly on the alterations in Ascomycota and Basidiomycota. By secreting a range of cellulases, rumen fungi primarily work to further degrade the components of fibrous tissue and to destroy structural carbohydrates. Rumen bacteria’s capacity to attach to fibers and the quantity of Ascomycota were both impacted. This was in line with Pitta’s observation that the presence of Ascomycota was influenced by fiber content [31].

At the fungal genus level, the dominant fungi were *Thelebolus*, *Cladosporium*, and *Meyerozyma*. Although there was no significant change in *Thelebolus* and *Cladosporium* at each incubated time, but the number of *Meyerozyma* in CORNSTR showed an obvious increase trend. It was speculated that the fiber structure of corn straw was more conducive to the attachment, growth and reproduction of *Meyerozyma*. In addition, with the increase in fiber content in roughage, Cellulase enzyme activity in rumen was activated, which was also responsible for the increase in the number of yeast-like fungi. The specific reasons need further study. These results reflect those of Fliegerova et al., who found that the different diet compositions might not reflect similarities in the adaptation of rumen microbial populations but to the rapid digestion rate of dietary soluble carbohydrate and protein substrates [11]. In summary, these results show that the rumen of beef cattle fed with straw roughages often contained more fiber-decomposing bacteria and free enzyme proteins involved in fiber decomposing, especially fungus.

## 5. Conclusions

The COMB used as roughage in beef cattle diet can improve ruminal CP and DM degradability positively. However, this combination’s rate of NDF and ADF degradation is lower than that of CORNSTR. In addition, the three treatments of roughages had no significant effect on the abundance of the fungal community, but could still maintain a healthy fungal community and rumen environment. The *Meyerozyma* quantity in CORNSTR was the largest, which was helpful to further study the response of *Meyerozyma* to the change in corn straw diet with different ratios. It is worth experimenting with straw to control fibrolytic bacteria in the rumen. In conclusion, it is suggested to feed beef cattle with different proportions of millet straw and corn straw combinations.

## Figures and Tables

**Figure 1 animals-13-00548-f001:**
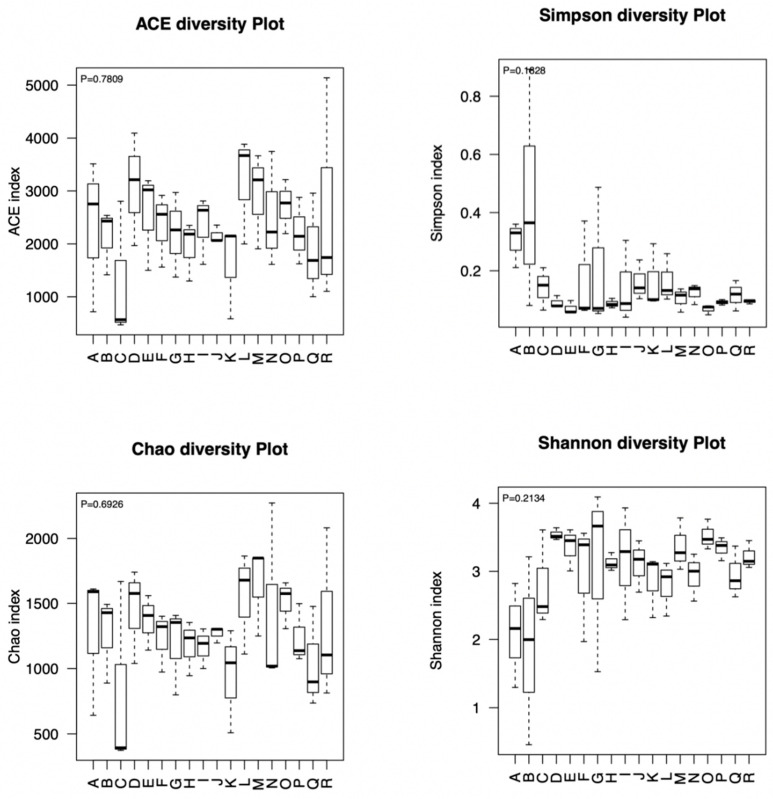
The fungus α-diversity in each treatment at different incubation times (A = MILLSTR 6 h; B = MILLSTR 12 h; C = MILLSTR 24 h; D = MILLSTR 36 h; E = MILLSTR 48 h; F = MILLSTR 72 h; G = COMB 6 h; H = COMB 12 h; I = COMB 24 h; J = COMB 36 h; K = COMB 48 h; L = COMB 72 h; M = CORNSTR 6 h; N = CORNSTR 12 h; O = CORNSTR 24 h; P = CORNSTR 36 h; Q = CORNSTR 48 h; R = CORNSTR 72 h; the same as below).

**Figure 2 animals-13-00548-f002:**
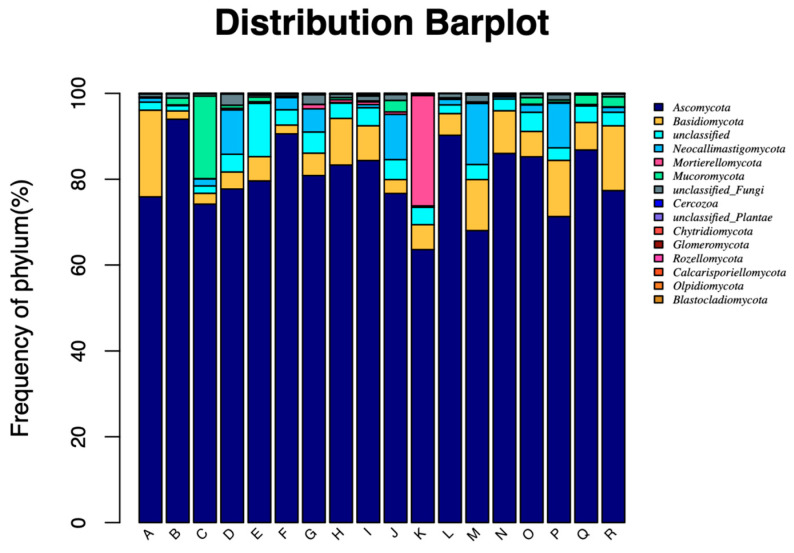
Phylum composition of fungus in each treatment at different incubation times.

**Figure 3 animals-13-00548-f003:**
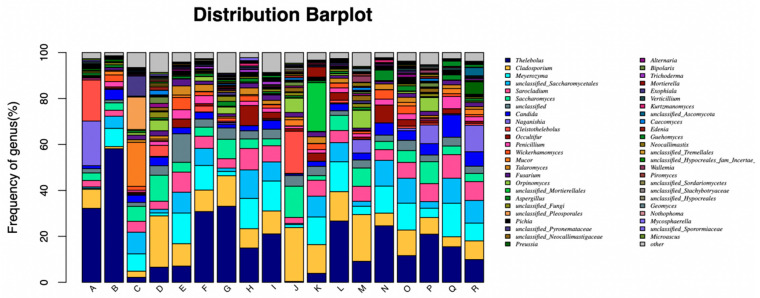
Genus composition of fungus in each treatment, at different incubation times.

**Table 1 animals-13-00548-t001:** Composition of the experimental diets.

Items	Groups		
	MILLSTR	COMB	CORNSTR
Ingredient			
Corn	35.22	35.22	35.22
Soybean meal	6.35	6.35	6.35
Wheat bran	3.7	3.7	3.7
Linseed cake meal	2	2	2
Premix ^1^	5	5	5
Salt	0.2	0.2	0.2
Sodium bicarbonate	0.42	0.42	0.42
Corn straw	0	23.56	47.11
Millet straw	47.11	23.56	0
Total	100	100	100
Nutrient composition			
Net energy (MJ/kg)	6.57	6.54	6.52
Crude protein (%)	13	12.5	11.4
Neutral detergent fiber (%)	31.8	30.5	30.5
Acid detergent fiber (%)	21.6	20.6	20.3
Calcium (%)	0.7	0.7	0.7

^1^ Premix content per kilogram: VA 160,000 IU, VD3 60,000 IU, VE 450 mg, Fe 500 mg, Mn 400 mg, Zn 500 mg, Cu 100 mg, I 13 mg, Se 5 mg, Co 5 mg, NaCl 10%.

**Table 2 animals-13-00548-t002:** Nutritional components of millet straw and corn straw (air-dry basis).

Ingredients	NE (MJ/kg)	CP (%)	DM (%)	Ca (%)	P (%)	NDF (%)	ADF (%)
Millet straw	3.58	5.9	95.19	0.25	0.24	76.24	42.19
Corn straw	2.74	3.8	95.1	0.32	0.21	80.01	43.54

NE = net energy; CP = crude protein; DM = dry matter; NDF = neutral detergent fiber; ADF = acid detergent fiber. The same as below.

**Table 3 animals-13-00548-t003:** Effect of different diets on dynamic pH at different incubation times.

Time Points	MILLSTR	COMB	CORNSTR
6 h	6.52 ± 0.43	6.34 ± 0.44	6.70 ± 0.31
12 h	7.25 ± 0.09	7.12 ± 0.27	7.22 ± 0.17
24 h	7.06 ± 0.22	7.01 ± 0.28	7.09 ± 0.13
36 h	7.05 ± 0.07	6.94 ± 0.16	6.94 ± 0.05
48 h	7.24 ± 0.16	7.20 ± 0.31	7.03 ± 0.17
72 h	6.93 ± 0.30	6.72 ± 0.48	6.85 ± 0.27

In the same row, values with no letter or the same letter superscripts mean no significant difference (*p* > 0.05), with different small-letter superscripts meaning significant difference (*p* < 0.05). The same as below.

**Table 4 animals-13-00548-t004:** Effect of different diets on the DM degradability at different incubation times.

Items	MILLSTR	COMB	CORNSTR
6 h	21.18 ± 0.70	23.17 ± 2.30	21.44 ± 0.20
12 h	25.84 ± 2.26 ^bc^	30.43 ± 0.99 ^a^	26.21 ± 0.96 ^b^
24 h	35.90 ± 1.13 ^bc^	39.74 ± 1.01 ^a^	36.68 ± 0.13 ^b^
36 h	45.48 ± 4.09	47.34 ± 4.49	46.45 ± 3.59
48 h	50.71 ± 3.00	54.76 ± 3.03	52.43 ± 0.97
72 h	54.23 ± 0.96 ^c^	59.87 ± 2.02 ^a^	56.97 ± 3.81 ^ab^
a	3.0 ± 0.27	1.99 ± 0.79	2.40 ± 0.29
b	50.72 ± 1.73	51.97 ± 4.77	52.50 ± 1.82
c (%/h)	5.34 ± 0.35 ^bc^	6.74 ± 1.08 ^a^	5.23 ± 0.16 ^b^
ED	52.73 ± 1.88	53.71 ± 5.39	54.60 ± 1.53

^a^ was rapidly degraded proportion, ^b^ was slowly degraded proportion, and ^c^ was the degradation speed of ^b^; ED was effective degradation rate.

**Table 5 animals-13-00548-t005:** Effect of different diets on the CP degradability at different incubation times.

Items	MILLSTR	COMB	CORNSTR
6 h	34.20 ± 5.75 ^b^	45.87 ± 1.65 ^a^	23.11 ± 0.54 ^c^
12 h	44.94 ± 1.00 ^b^	53.53 ± 0.40 ^a^	28.23 ± 0.87 ^c^
24 h	58.16 ± 0.39 ^b^	60.53 ± 0.75 ^a^	39.70 ± 0.42 ^c^
36 h	66.70 ± 0.49 ^ab^	67.37 ± 2.76 ^a^	50.68 ± 3.36 ^c^
48 h	75.65 ± 0.17 ^ab^	76.33 ± 1.65 ^a^	59.80 ± 1.13 ^c^
72 h	86.20 ± 1.67 ^ab^	87.48 ± 0.98 ^a^	67.76 ± 3.00 ^c^
a/%	2.03 ± 0.76 ^b^	1.18 ± 0.34 ^bc^	3.17 ± 0.22 ^a^
b/%	70.73 ± 2.58 ^a^	67.65 ± 1.64 ^ab^	62.00 ± 1.04 ^c^
c (%/h)	8.17 ± 1.23 ^b^	14.85 ± 1.24 ^a^	4.33 ± 0.09 ^c^
ED/%	72.49 ± 1.82 a	68.69 ± 1.97 ^b^	64.73 ± 1.03 ^c^

^a^ was rapidly degraded proportion, ^b^ was slowly degraded proportion, and ^c^ was the degradation speed of ^b^; ED was effective degradation rate.

**Table 6 animals-13-00548-t006:** Effect of different diets on the NDF degradability at different incubation times.

Items	MILLSTR	COMB	CORNSTR
6 h	15.07 ± 1.48	15.34 ± 1.59	17.68 ± 0.77
12 h	20.02 ± 1.98	20.50 ± 1.92	21.86 ± 1.66
24 h	29.27 ± 1.68 ^c^	31.43 ± 0.94 ^bc^	32.24 ± 0.35 ^ab^
36 h	39.99 ± 4.36	38.62 ± 3.21	42.83 ± 4.36
48 h	45.48 ± 3.55	46.59 ± 1.76	49.55 ± 0.40
72 h	51.30 ± 0.71 ^c^	55.18 ± 2.27 ^bc^	58.63 ± 4.47 ^ab^
a/%	1.89 ± 0.94	1.71 ± 0.56	2.36 ± 0.44
b/%	53.30 ± 2.35	52.17 ± 6.38	55.81 ± 3.92
c (%/h)	3.43 ± 0.23	3.86 ± 0.85	3.67 ± 0.42
ED/%	54.71 ± 2.42	53.44 ± 5.89	57.70 ± 3.45

^a^ was rapidly degraded proportion, ^b^ was slowly degraded proportion, and ^c^ was the degradation speed of ^b^; ED was effective degradation rate.

**Table 7 animals-13-00548-t007:** Effect of different diets on the ADF degradability at different incubation times.

Items	MILLSTR	COMB	CORNSTR
6 h	9.31 ± 0.92 ^c^	12.07 ± 0.54 ^b^	14.26 ± 1.22 ^a^
12 h	15.59 ± 0.65	16.82 ± 0.62	17.68 ± 1.97
24 h	23.68 ± 2.51 ^c^	26.22 ± 0.53 ^ab^	26.82 ± 0.55 ^a^
36 h	33.84 ± 4.12	34.28 ± 2.13	36.01 ± 4.93
48 h	39.75 ± 3.89 ^c^	44.58 ± 1.95 ^bc^	45.84 ± 0.27 ^ab^
72 h	45.88 ± 1.50 ^c^	51.75 ± 3.40 ^bc^	54.77 ± 5.17 ^ab^
a/%	1.41 ± 0.45 ^c^	1.99 ± 0.06 ^bc^	2.67 ± 0.92 ^ab^
b/%	62.15 ± 13.07	70.00 ± 11.63	70.34 ± 9.86
c (%/h)	2.21 ± 0.76	1.91 ± 0.38	1.98 ± 0.58
ED/%	62.60 ± 12.24	70.83 ± 11.24	71.84 ± 9.87

^a^ was rapidly degraded proportion, ^b^ was slowly degraded proportion, and ^c^ was the degradation speed of ^b^; ED was effective degradation rate.

**Table 8 animals-13-00548-t008:** Relative abundance of top 2 species at the phylum level (%).

Time Points	*Ascomycota*			*Basidiomycota*		
	MILLSTR	COMB	CORNSTR	MILLSTR	COMB	CORNSTR
6 h	77.60 ± 25.82	93.60 ± 3.83	73.91 ± 15.89	18.17 ± 29.31	2.02 ± 1.49	2.51 ± 2.53
12 h	77.88 ± 4.58	79.46 ± 14.06	90.37 ± 7.26	3.89 ± 1.48 ^ab^	5.66 ± 0.75 ^a^	2.03 ± 0.69 ^b^
24 h	78.86 ± 15.55	83.98 ± 4.24	84.06 ± 4.62	5.82 ± 4.73	10.11 ± 4.87	8.12 ± 1.09
36 h	76.01 ± 5.67	67.68 ± 32.39	90.33 ± 5.50	3.06 ± 1.30	5.93 ± 1.30	4.91 ± 3.65
48 h	65.68 ± 16.40	83.71 ± 12.03	85.17 ± 3.77	12.97 ± 11.84	11.92 ± 10.24	6.10 ± 2.64
72 h	71.11 ± 9.84	86.63 ± 4.84	78.23 ± 11.86	13.38 ± 8.08	6.75 ± 5.12	14.50 ± 11.88

In the same row, values with no letter or the same letter superscripts mean no significant difference (*p* > 0.05), with different small-letter superscripts meaning significant difference (*p* < 0.05). The same as below. ^a^ was rapidly degraded proportion; ^b^ was slowly degraded proportion.

**Table 9 animals-13-00548-t009:** Relative abundance of top 3 species at the genus level (%).

Time Points	*Thelebolus*			*Cladosporium*			*Meyerozyma*		
	MILLSTR	COMB	CORNSTR	MILLSTR	COMB	CORNSTR	MILLSTR	COMB	CORNSTR
6 h	33.78 ± 30.21	54.81 ± 42.72	2.19 ± 3.62	8.86 ± 7.66	1.00 ± 1.37	2.65 ± 2.68	0.45 ± 0.38	8.64 ± 7.84	7.5 ± 3.07
12 h	6.80 ± 8.61	7.18 ± 9.86	26.51 ± 30.09	21.90 ± 9.02	9.62 ± 5.49	10.03 ± 5.19	1.30 ± 0.34 ^c^	13.42 ± 5.09 ^a^	10.41 ± 1.55 ^ab^
24 h	28.78 ± 35.63	13.0 ± 12.40	18.4 ± 30.69	14.10 ± 9.76	8.52 ± 3.92	10.6 ± 5.77	2.09 ± 3.25 ^c^	13.78 ± 4.27 ^a^	13.51 ± 4.23 ^ab^
36 h	0.37 ± 0.18	4.22 ± 4.50	26.02 ± 24.48	23.56 ± 6.73	13.22 ± 13.21	12.67 ± 12.22	1.35 ± 0.78 ^c^	13.07 ± 8.56 ^a^	13.04 ± 1.15 ^ab^
48 h	11.02 ± 10.43	21.09 ± 18.32	10.3 ± 8.68	19.30 ± 9.37	6.85 ± 6.98	11.91 ± 9.37	2.82 ± 3.69 ^c^	11.45 ± 6.22 ^bc^	11.87 ± 2.00 ^ab^
72 h	18.99 ± 14.55	14.33 ± 17.20	9.37 ± 16.05	7.06 ± 3.39	4.60 ± 3.69	9.64 ± 13.24	4.74 ± 5.32 ^c^	14.54 ± 3.93 ^ab^	7.88 ± 1.37 ^bc^

^a^ was rapidly degraded proportion, ^b^ was slowly degraded proportion, and ^c^ was the degradation speed of ^b^; ED was effective degradation rate.

## Data Availability

All data are publicly available.

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
