# Peer review of "The Effect of Combining Millet and Corn Straw as Source Forage for Beef Cattle Diets on Ruminal Degradability and Fungal Community"

_animals, 2023, doi:10.3390/ani13040548_

Round 1

Reviewer 1 Report

The suggestion of this manuscript were:

L47-81, Introduction should add more detail about role of roughage on rumen fungal community.

L25, Materials and methods: This experiment study about roughage source which take long time to degraded, why study only 72h why not study for 96 h.

L173: In status ical analysis, why this experiment don't use CRD.

L306: Add number of reference of "Fliegerova et al. ().

Author Response

Hello and thanks for the wise review.I have revised the article according to your suggestion.I need to state that the experimental results of 96h were not measured in the original design of the experimental scheme, and CRD was not used in the analysis method.If there is anything that needs to be modified, I will correct it.

Reviewer 2 Report

The fact that the study examined the effects of different diets on rumen microbial changes and their degradation, particularly focusing on fungi, but there have been similar results in the past, so what makes this study different from the others?

 It would be good to include it more clearly in the introduction and in the summary of the results.

1) Please write the official names for DM, CP, NDF, and ADF when these words first appear.

2) The way the results written was too simple. Since there were much data of microbial communities, it would be better to include statistical analysis such as Principal Coordinate Analysis to show how the overall structure of the microbial community is changing, and to summarize the changes in the community as a whole statistically and in a way that is easy to understand.

3) There are other papers on fungal dynamics in the rumen, such as the one below.
“An Investigation into Rumen Fungal and Protozoal Diversity in Three Rumen Fractions, during High-Fiber or Grain-Induced Sub-Acute Ruminal Acidosis Conditions, with or without Active Dry Yeast Supplementation”
Front. Microbiol., 10 October 2017
Sec. Microbial Symbioses 
https://doi.org/10.3389/fmicb.2017.01943

It would be better to cite other previous studies and discuss the results in more detail and depth.

4)Please write what is known and discuss Meyerozyma, citing previous studies.

5) There is a list of 56 references, but not all references were used in the actual text, and the numbers they appear in are scattered and need to be organized.

Author Response

Hello and thanks for the wise review.I have revised the article according to your suggestion.If there is anything that needs to be modified, I will correct it.

Reviewer 3 Report

Title 

The high-forage concept is not clear, I suggest changing to “The effect of combining millet and corn stover as source forage for beef cattle diets on ruminal degradability and fungal community”

Abstract

L31 and L80 specify what was determined through sequence analysis of 18S rDNA and indicate if it was in the complete, liquid or solid fraction of the rumen content.

Materials and Methods

Table 1. 

Change fifiber to fiber

Indicate at the bottom of the table all the acronyms Nemf

homogenize with table 2 if acronyms or full names are used for crude protein, neutral detergent fiber etc.

Table 2. Indicate at the bottom of the table all the acronyms Nemf, CP, DM, etc.

Include lignin content

Discussion

L274-277. Due to differences in lignin content, which can help explain differences in the rumen degradability of NDF and ADF, it is important to analyze it and present the results.

L292-293. As you indicate, "Fungus adhere to segments of roughage in the rumen of ruminants, especially when fed diets high in fiber", you should discuss whether you used the liquid fraction of the rumen, and why not the solid or complete fraction. And if you expect differences from what the literature indicates.

L295 Discuss why Basidiomycota in COMB was higher than in CORNSTR at 12 h?

L306 Fligerova et la (year??)

Author Response

Hello and thanks for the wise review.I have revised the article according to your suggestion. If there is anything that needs to be modified, I will correct it.

Round 2

Reviewer 1 Report

Interesting work but if discuss more about who do before and how happen same or difference from this result.

Reviewer 2 Report

References are only cited up to number 30, but the list goes up to number 57, so Delete references not used in the text.

In line 299, According to table 8, the number of Basidiomycota in COMB was higher (P<0.05) than that in CORNSTR at 12h, and there was no significant difference (P >0.05) at other incubation times, which seemed to indicate that adaptation of the rumen fungal community to dietary changes, shows the rate of presence per total community, which is strictly It is not a NUMBER in the strict sense.

If we know something such as the overall amount of each community is the same, the percentage can be converted to a quantity, but if we are not measuring the overall amount of microorganisms, shouldn't we still discuss it as a percentage?

At the genus level of fungi, names are written in italics. Please correct the genus level names in the text and tables, as they are not italicized.

Reviewer 3 Report

Points 4, 5 and 6 were not answered

They answer in all cases were: "The article has been revised"

But no changes were made

Points 4. Table 2. Indicate at the bottom of the table all the acronyms Nemf, CP, DM, etc.

Include lignin content

Discussion

Point 5.  L274-277. Due to differences in lignin content, which can help explain differences in the rumen degradability of NDF and ADF, it is important to analyze it and present the results.

Point 6. L292-293. As you indicate, "Fungus adhere to segments of roughage in the rumen of ruminants, especially when fed diets high in fiber", you should discuss whether you used the liquid fraction of the rumen, and why not the solid or complete fraction. And if you expect differences from what the literature indicates.
